# Improved Active Disturbance Rejection Control of Dual-Axis Servo Tracking Turntable with Friction Observer

**Qian Zhang** [1,2] , **Xu Wu** [2,3], **Qunjing Wang** [2,*], **Dijiang Chen** [4] **and Chao Ye** [5]

1   School of Electrical Engineering and Automation, Anhui University, Hefei 230601, China; qianzh@ahu.edu.cn
2   National Engineering Laboratory of Energy-Saving Motor & Control Technology, Anhui University, Hefei 230601, China; z19201001@stu.ahu.edu.cn
3   Research Center of Power Quality, Ministry of Education, Anhui University, Hefei 230601, China
4   Anhui Key Laboratory of Industrial Energy-Saving and Safety, Anhui University, Hefei 230601, China; chendij@163.com
5   The 38th Research Institute of China Electronics Technology Group Corporation, Hefei 230088, China; yechao202107@163.com
*   Correspondence: wangqunjing@ahu.edu.cn

**Abstract:** Friction nonlinear disturbance is one of the main factors affecting the control performance of servo tracking system. In this paper, an improved Active Disturbance Rejection Control (ADRC) scheme of dual-axis servo turntable is researched to achieve accurate tracking. Firstly, the mathematical dynamics model of dual-axis servo tracking turntable system is established. The Elastoplastic model is used to describe nonlinear friction, in which the immeasurable part is extended to be a new state. Secondly, considering the smooth and monotonic increasing property of hyperbolic tangent function, an improved tracking differentiator is introduced, which can provide better noise attenuation performance. Thirdly, based on adjustable parameter systematic pole placement method, the fuzzy control algorithm is applied to realize the intelligent tuning of the improved Extended State Observer (ESO) gains, in which the input of the fuzzy controller is the estimation error, while the output is the observer bandwidth. Finally, the improved ADRC system is transformed into a Lurie system, then the extended circle criteria are adopted to analyze the absolute stability of the proposed system. Simulation and experimental verification of the improved ADRC method for the dual-axis turntable tracking servo system are conducted. Results illustrate the effectiveness and robustness of the proposed controller.

**Keywords:** motor-drive servo system; ADRC; tracking control; friction compensation

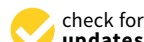



## 1. Introduction

The motor-drive servo turntable is widely used in high-precision tracking radars [1], radio telescopes [2], inertial navigation systems [3], and other equipment that require high tracking accuracy. It is of great importance to design a controller for turntable servo system with accurate tracking performance. In recent years, extensive research on the electromechanical servo systems have been conducted locally and internationally [4–6]. Some advanced control strategies, such as robust method [7], adaptive control [8,9], model predictive method [10,11], sliding control [12,13], and back-stepping control [14], etc. are often employed together to acquire a better control performance. In order to describe the characteristics of servo turntable system in different working conditions, the switched theory based on Constrained Multi-objective Optimization Problem (CMOP) is introduced in [15]. In this study, an improved Active Disturbance Rejection Controller (ADRC), mainly given by an improved Tracking Differentiator (TD) and the improved Extended State Observer (ESO), is analyzed in detail.

The ADRC, proposed by Han [16], as a nonlinear control method with superior performance, are preferred for a wide application in industrial technical fields [17–21].

In [17], an adaptive linear ADRC is designed to acquire strong disturbance rejection performance and to reduce the noise sensitivity for electromechanical servo system. In [18], it has been proved that the ADRC is with stronger anti-interference ability than PID, Fuzzy-PID, and BP-PID under the condition of 20% maximum control torque disturbance. To solve the output tracking problem of multi-input multi-output (MIMO) system with mismatched uncertainty effectively, a novel nonlinear ESO using nonsmoothed function [19] is designed to estimate both state of system and uncertain disturbance. In [20], the influence of input-gain uncertainty on tracking performance of ADRC is investigated based on a second-order plant. The results show that when the input gain is multiplied by a positive factor less than a certain threshold, the closed-loop system remains stable and the tracking error is reduced. For the purpose of improving the tracking accuracy of servo system effectively, an ADRC and feedback linearization-based control algorithm for the high-precision trajectory tracking is analyzed in [21].

The turntable servo system often fails to achieve accurate tracking due to nonlinear disturbance factors such as friction [22–26], backlash [27], dead-zone [28], and motor torque fluctuation [29], etc. Friction is a common non-linear phenomenon, which exists in almost all electromechanical servo system between two contact surfaces with relative motion. Among the existing nonlinear friction models, the LuGre friction model [22,23] is most widely employed to describe nonlinear friction phenomenon because of its ability to capture most of the observed frictional behaviors. Nevertheless, some drifting behavior inevitably exists in the LuGre friction model. To deal with this challenge effectively, another single state friction model was proposed by Dupont in 2002, which is called the Elastoplastic (EP) friction model [24]. It could be concluded that the EP friction model can overcome this drawback by incorporating a purely elastic area. In [26], the dynamic parameters of EP friction model are identified and used for feed-forward compensation based on experimental measurement results.

However, it should not be ignored that most of the existing friction models are based on the deformation of rigid bristles, while the average deformation of rigid bristles is often too small to be measured. In this paper, to describe nonlinear friction disturbance more accurately, we consider regarding the immeasurable part of the EP friction model as a new state variable to constitute a novel ESO, which is the core of ADRC controller design.

It is not difficult to find out that the ESO used in the aforementioned studies were generally designed on the basis of traditional nonlinear function, e.g., fal. Although they are continuous, it is hard to guarantee the derivability on the intervals, and they have an inflection point around the origin, which may affect the control performance of ADRC. As a result, the ADRC controller can be improved by optimizing its structure, which is introduced in Section 3.

Stability provides a prerequisite for the normal operation of turntable servo system. In [30], the stability of traditional PID controller is investigated, then the extended PID is proposed to acquire more strong robustness. Results show that the extended PID controller is able to stabilize the nonlinear uncertain systems semi globally. In [31], for the purpose of improving the tracking performance and robustness of robotic manipulators, the neural network (NN) algorithm is introduced to model predictive control (MPC). We have learned from previous studies that nonlinear ESO can obtain better performance than linear ESO [32–34]. However, the stability analysis of nonlinear ESO is a challenging work. Most of the existing methods, such as circle criterion method [35], describing function method [36], are carried out in frequency domain. In this paper, the improved ADRC system is transformed into a Lurie system, then the extended circle criterion is employed to analyze the stability in frequency domain.

The main contributions of this study are as follows: (1) the immeasurable part of the Elastoplastic friction model is extended to a new state to achieve real-time estimation and compensation; (2) the fuzzy rules are introduced to realize intelligent tuning of the improved ESO gains based on the adjustable parameter systematic pole placement method;

and (3) the designed control system above is transformed into Lurie system, then the extended circle criterion is adopted to analyze stability of the improved ADRC system.

The remaining part of this paper is organized as follows. The dynamic model of dual-axis servo tracking turntable system is shown in Section 2. In Section 3, the improved ADRC is designed based on improved TD and improved ESO. The gains of improved ESO are adjusted intelligently by fuzzy rules. In Section 4, the improved ADRC system above is transformed into a Lurie system, then the stability of designed system is analyzed by extended circle criterion. Simulation analysis and experimental results are given in Sections 5 and 6, respectively. The results illustrate that the improved ADRC control scheme can improve the tracking performance of dual-axis servo turntable system. Finally, some concluding remarks are drawn in Section 7.

## 2. Dynamic Model of Dual-Axis Servo Tracking Turntable System

The dual-axis servo turntable tracking system driven by servo motor is the controlled plant as shown in Figure 1, which is mainly composed of servo motor, integrated motor driver, PWM module, RTU-BOX real-time digital controller, host computer, and transmission mechanism, such as worm and gear, etc.

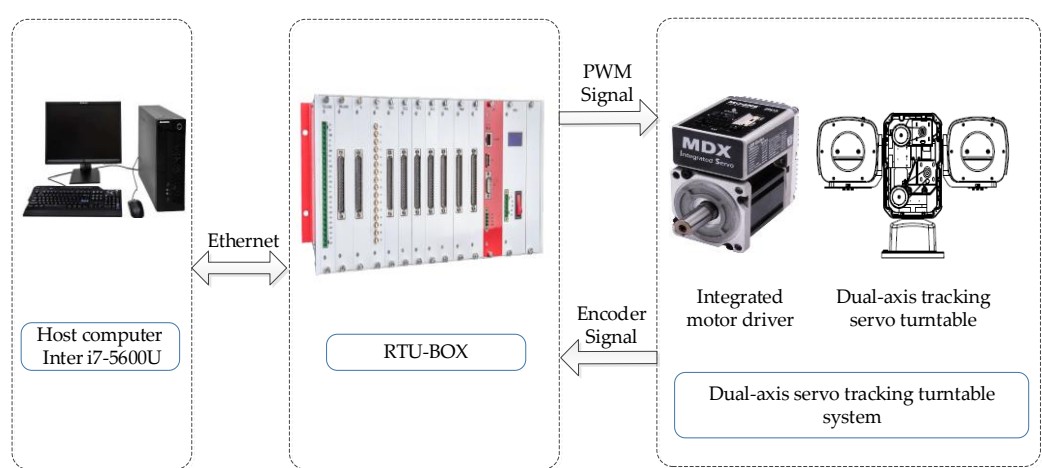

**Figure 1.** Diagram of dual-axis servo tracking turntable system.

*2.1. Mechanism Modelling of Servo Turntable*

In this study, $\theta$ represents the output angle of servo tracking turntable. The dynamic voltage equation of drive servo motor can be described as follows

$$U_a = I_a R_a + E_a + L_a \frac{dI_a}{dt} \tag{1}$$

where $U_a$ and $I_a$ are the armature voltage and current of the servo motor separately, $R_a$ and $L_a$ are armature resistance and inductance, and $E_a$ is the back electromotive force defined as $E_a = K_e \dot{\theta}$.

The torque balance equation is given as

$$T_e = J \frac{d^2\theta}{dt^2} + T_f \tag{2}$$

where $J$ is the moment of inertia of servo motor and $T_e$ is the electromagnetic torque defined as $T_e = K_t I_a$, $T_f$ is friction torque.

Ignoring the influence of friction torque, the transfer function model of servo system can be obtained by Laplace transform of the above formula under zero initial condition as follows

$$N \frac{\dot{\Theta}(s)}{U_a(s)} = \frac{N \cdot K_t}{J L_a s^2 + J R_a s + K_t K_e} = \frac{\frac{N}{K_e}}{\tau_1 \tau_2 s^2 + \tau_1 s + 1} \tag{3}$$

where $\tau_1$ and $\tau_2$ are the electromechanical time constant and electromagnetic time constant denoted as $\tau_1 = \frac{JR_a}{K_t K_e}$, $\tau_2 = \frac{L_a}{R_a}$, respectively, and $N$ is the transmission ratio of tracking turntable to servo motor.

### 2.2. Nonlinear Friction Modelling

The Elastoplastic (EP) friction model is adopted to describe nonlinear friction disturbance [24–26], which is described by the average deflection of bristles as follows

$$\dot{z} = \dot{\theta}\left[1 - \alpha\left(z, \dot{\theta}\right)\frac{z}{z_{ss}\left(\dot{\theta}\right)}\right] \tag{4}$$

where $z$ is the average deformation of rigid bristles, which is an immeasurable internal friction state. $\dot{\theta}$ is output angular velocity of the servo turntable and $z_{ss}\left(\dot{\theta}\right)$ is a monotonically decreasing function described by Stribeck effect as follows

$$\sigma_0 z_{ss}\left(\dot{\theta}\right) = T_c + (T_s - T_c)e^{-\left(\frac{\dot{\theta}}{v_s}\right)^2} \tag{5}$$

where $\sigma_0$ is stiffness coefficient of bristles, $v_s$ is the Stribeck angular velocity, and $T_c$ and $T_s$ are Coulomb friction torque and maximum static friction torque, respectively. The remaining parameters of EP friction model are defined as below

$$\alpha\left(z, \dot{\theta}\right) = \frac{1}{2}\sin\left(\pi\frac{z - 0.5(z_{ss} + z_{ba})}{z_{ss} - z_{ba}}\right) + \frac{1}{2} \tag{6}$$

The above formula is given under condition of $z_{ba} < |z| < z_{ss}$, where $z_{ba} = 0.7169 \cdot z_{ss}$, then the friction torque of servo turntable can be calculated as follows

$$T_f = \sigma_0 z + \sigma_1 \dot{z} + \sigma_2 \dot{\theta} \tag{7}$$

where $\sigma_1$ and $\sigma_2$ are the micro-damping and viscous friction coefficients, respectively.

### 3. Design of Improved Active Disturbance Rejection Controller

From prior knowledge, the active disturbance rejection controller (ADRC) is mainly composed of tracking differentiator (TD), extended state observer (ESO), and state error feedback (SEF) [16]. In this section, an improved tracking differentiator based on hyperbolic tangent function is given. The improved extended state observer using a new nonlinear function is discussed and analyzed in detail, and the gains of ESO are tuned by observer bandwidth adopting fuzzy rules. The structure diagram of improved ADRC is shown in Figure 2.

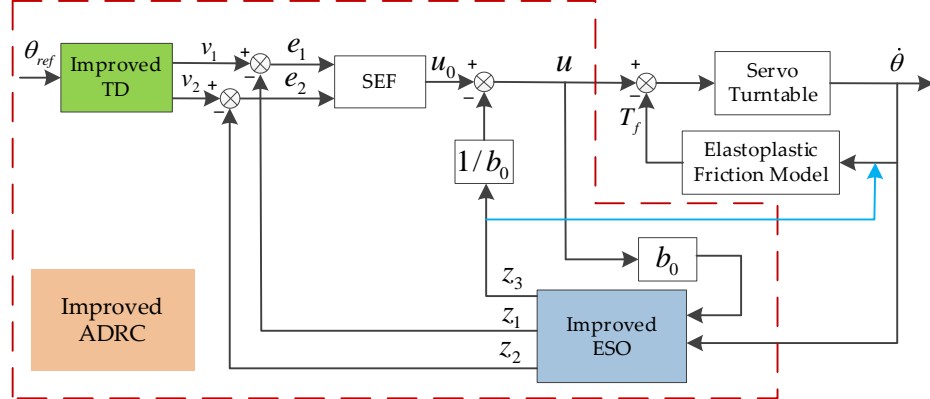

**Figure 2.** The structure diagram of improved ADRC.

### 3.1. Improved Tracking Differentiator

Before further research, the properties of hyperbolic tangent function, shown in [32], are worth analyzing first. It can be approximated to a linear term in some sufficiently small interval, with features of smooth and monotonically increasing at the same time. That makes hyperbolic tangent function an appropriate choice for designing TD.

The improved TD with n-order based on hyperbolic tangent function is given as follows

$$\begin{cases} \dot{v}_1(t) = v_2(t) \\ \cdots \\ \dot{v}_{n-1}(t) = v_n(t) \\ \dot{v}_n(t) = R^n \cdot tanh\left[v_1(t) - v(t), \frac{v_2(t)}{R}, \cdots, \frac{v_n(t)}{R^{n-1}}\right] \end{cases} \tag{8}$$

where $v_i(t)$, $i = 2, \ldots, n$ are differential of each order of input signal $v(t)$ and $v_1(t)$ is the approximate estimation of $v(t)$.

TD is generally adopted to arrange transition process reasonably. When servo system switches working conditions frequently, a transition process is necessary to reduce the unexpected impact of trembling on system. In this section, the square wave input is chosen as the testing signal to verify effectiveness of improved TD, which is shown in Figures 3 and 4, respectively. The square wave tracking employing ordinary differentiator is with 15.6% overshot, while it is almost zero for improved TD. Obviously, the improved TD can arrange transition process more properly than ordinary differentiator to extract differential signal.

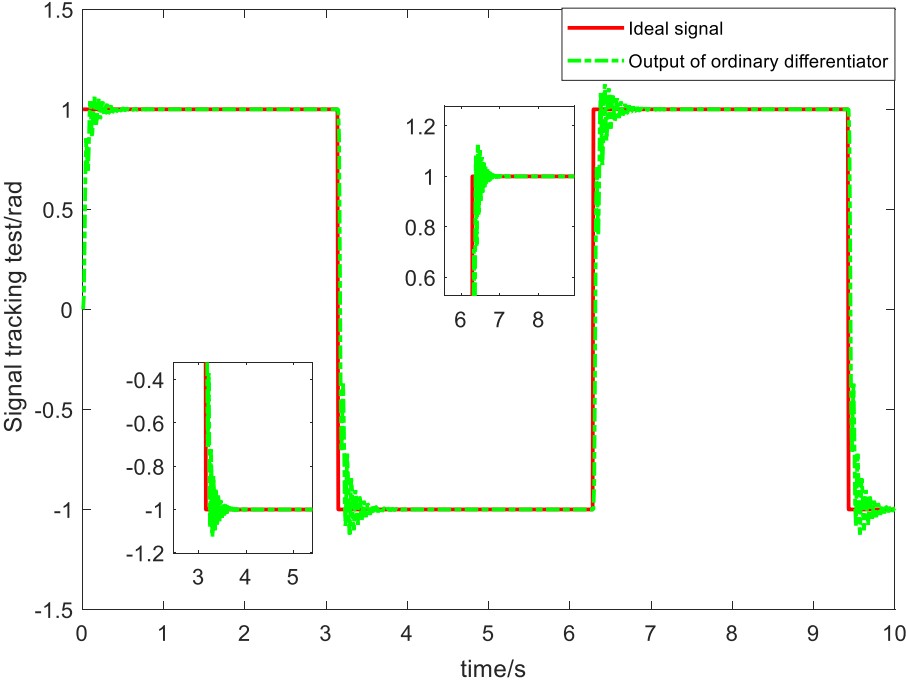

**Figure 3.** Square wave tracking based on ordinary differentiator.

In this study, considering the servo turntable plant is a second order system, then Equation (8) is simplified as follows

$$\begin{cases} \dot{v}_1 = v_2 \\ \dot{v}_2 = R^2\left[-m \cdot \tan h(p(v_1 - v)) - n \cdot \tan h\left(q\frac{v_2}{R}\right)\right] \end{cases} \tag{9}$$

where $R > 1$, $0 < m, n, p, q \leq 1$ are the positive adjustable parameters, and the time variable is omitted for brevity.

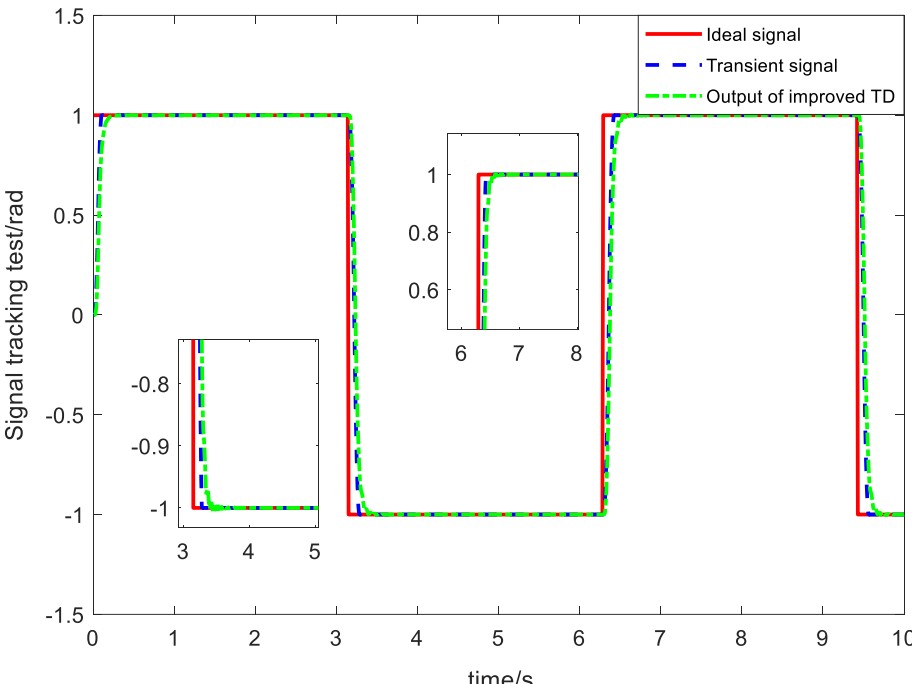

**Figure 4.** Square wave tracking based on improved TD.

### 3.2. Improved Extended State Observer

The ESO is mainly used to estimate and compensate the nonlinear disturbance of dual-axis turntable servo system, which is the core and essence of the ADRC. It is not hard to find out that the nonlinear ESO (NESO) can acquire a more superior performance than the linear ESO (LESO), and the design of NESO is almost based on the traditional nonlinear function, which is described as follows

$$fal(e, \alpha, \delta) = \begin{cases} |e|^{\alpha} \cdot sign(e), |e| > \delta \\ \frac{e}{\delta^{1-\alpha}}, |e| \leq \delta \end{cases} \tag{10}$$

However, there exists some inflection points near the origin, and it cannot guarantee to be differentiable in certain intervals. Consequently, the traditional function is supposed to be optimized. A new nonlinear function proposed for the design of improved ESO is shown in [33], which can be improved further.

Considering that in the interval of $|e| \leq \delta$, $\sin(x)$ is with better stationarity than $x$, and $\tan(x)$ is with better convergence than $x^3$, then a novel nonlinear function is given as follows

$$\varphi_i(e, \alpha, \delta, \gamma) = \begin{cases} (\alpha - 1)\delta^{\alpha-3}\tan(e) - (\alpha - 1)\delta^{\alpha-2}e^2 sign(e) + \frac{\sin(e)}{\delta^{1-\alpha}}, |e| \leq \delta \\ |e|^{\alpha} \cdot sign(e), \delta < |e| < \gamma \\ \gamma^{\alpha-1}, |e| \geq \gamma \end{cases} \tag{11}$$

then the improved ESO with n + 1-order based on new nonlinear function is expressed as follows

$$\begin{cases} e_1 = z_1 - \theta \\ \dot{z}_1 = z_2 - \beta_1 \varphi_1(e_1) \\ \cdots \\ \dot{z}_n = z_{n+1} - \beta_n \varphi_n(e_1) + b_0 u \\ \dot{z}_{n+1} = -\beta_{n+1} \varphi_{n+1}(e_1) \end{cases} \tag{12}$$

where $\beta_i$, $i = 1, \ldots, n + 1$ are the observer gains.

To choose the suitable $\beta_1, \cdots, \beta_{n+1}$, a LESO using systematic pole placement method is introduced first, which is described as follow

$$\begin{cases} e_1 = z_1 - \theta \\ \dot{z}_1 = z_2 - \beta_1 e_1 \\ \cdots \\ \dot{z}_n = z_{n+1} - \beta_n e_1 + b_0 u \\ \dot{z}_{n+1} = -\beta_{n+1} e_1 \end{cases} \tag{13}$$

$$(s + \omega_0)^{n+1} = s^{n+1} + \beta_1 s^n + \cdots + \beta_{n+1} \tag{14}$$

where $\omega_0$ is known as observer bandwidth. To acquire a better performance, the LESO generally applies larger gains, which may lead to peaking phenomenon. Accordingly, an appropriate selection of observer bandwidth is essential for achieving a reasonable trade-off between performance index and peaking value.

In this paper, an adjustable parameter systematic pole placement method is recommended to prevent the observer gains from being too large. The third order ESO used in this study is simplified from Equation (12) as follows

$$\begin{cases} e_1 = z_1 - \theta \\ \dot{z}_1 = z_2 - \beta_1 \varphi_1(e_1, \alpha_1, \delta, \gamma) \\ \dot{z}_2 = z_3 - \beta_2 \varphi_2(e_1, \alpha_2, \delta, \gamma) + b_0 u \\ \dot{z}_3 = -\beta_3 \varphi_3(e_1, \alpha_3, \delta, \gamma) \end{cases} \tag{15}$$

where $\beta_1 = 3\omega_0$, $\beta_2 = 3\omega_0^2/l_1$, $\beta_3 = \omega_0^3/l_2$, and $l_1$ and $l_2$ are the adjustable positive parameters. $z_1$, $z_2$, and $z_3$ are the estimated values of $\theta$, $\dot{\theta}$ and the immeasurable part of EP friction model, respectively, in this study.

Since the traditional tuning process of observer gains often depends on experience of industrial applications, there exists certain limitation. In order to make it intelligently, a fuzzy algorithm is employed as an appropriate tool in tuning observer gains. The sum of angular and angular velocity estimation error of NESO is regarded as the input of fuzzy controller, while the output is observer bandwidth. In this design, six membership indexes denoted as negative big (NB), negative medium (NM), negative small (NS), positive small (PS), positive medium (PM), and positive big (PB) are assigned for the input, within a $-0.8$ to 0.8 range. Three membership indexes named as small (S), medium (M), and big (B) are assigned for the output, within a 0 to 20 range. The membership function curves and the fuzzy algorithm output surfaces are given as Figures 5 and 6, respectively.

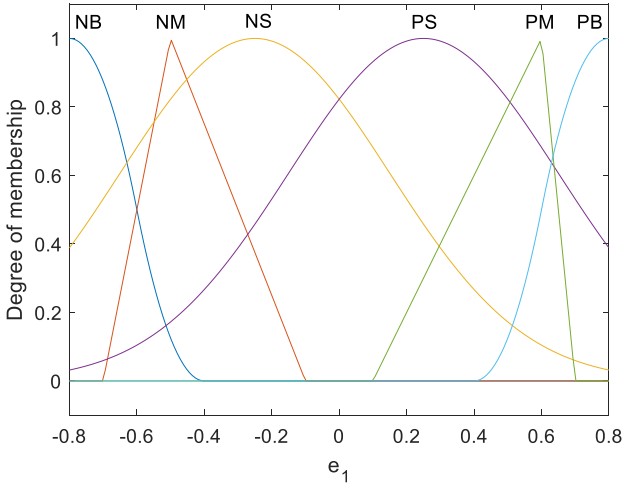

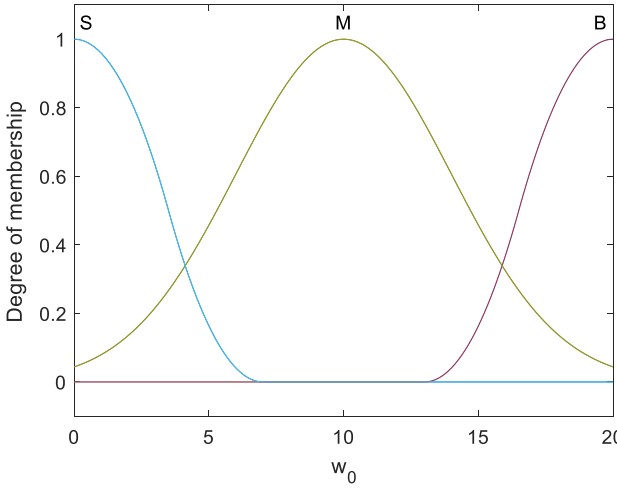

(**a**) Input member function curve

(**b**) Output member function curve

**Figure 5.** Input and output membership function curve.

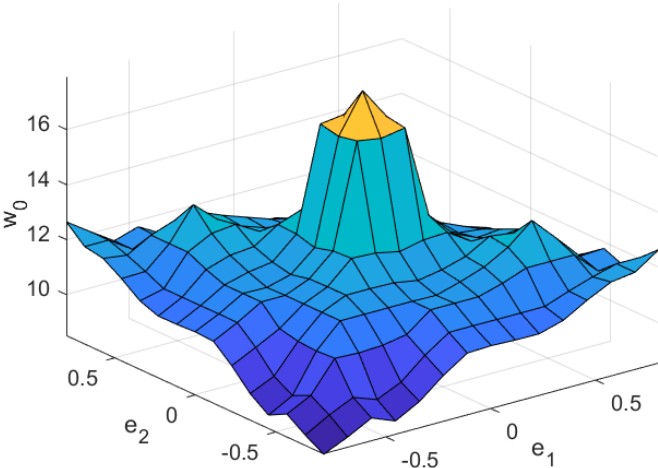

**Figure 6.** The fuzzy algorithm output surfaces.

The membership functions of NB and PB are adopted to cover some large input caused by initial disturbances and measurement noise. When the estimation error is large, the use of Z-shaped function can reduce the observer bandwidth in real time and then realize the intelligent adjustment of observer gains. The trigonometric and Gaussian function are applied for the medium and small values of estimation error, respectively.

After defining the membership function of input and output variables, the fuzzy rules are given as follows:

Rule 1: If $k_1 e_1 + k_2 e_2$ is NB, then $\omega_0$ is S.
Rule 2: If $k_1 e_1 + k_2 e_2$ is NM, then $\omega_0$ is M.
Rule 3: If $k_1 e_1 + k_2 e_2$ is NS, then $\omega_0$ is B.
Rule 4: If $k_1 e_1 + k_2 e_2$ is PS, then $\omega_0$ is B.
Rule 5: If $k_1 e_1 + k_2 e_2$ is PM, then $\omega_0$ is M.
Rule 6: If $k_1 e_1 + k_2 e_2$ is PB, then $\omega_0$ is S.

where $e_1$ is angle estimation error given in Equation (12), $e_2$ is the angle velocity estimation error defined as $e_2 = z_2 - \dot{\theta}$, $k_1$ and $k_2$ are the scaling factors.

Finally, in order to acquire a more accurate observer bandwidth output, the center of gravity defuzzifier is applied as follows by calculating the fuzzy output through Equation (16):

$$\omega_0^* = \frac{\int \omega_0 \cdot \mu(\omega_0) d\omega_0}{\int \mu(\omega_0) d\omega_0} \tag{16}$$

where $\mu(\omega_0)$ is the membership degree of $\omega_0$.

In the last part of this section, convergence analysis for the improved ESO is given. The general expression of nonlinear system with n+1-order can be described by the following equations:

$$\begin{cases} \dot{x}_1 = x_2 \\ \cdots \\ \dot{x}_n = x_{n+1} + b_0 u \\ \dot{x}_{n+1} = \psi(t) \\ y = \theta = x_1 \end{cases} \tag{17}$$

where $\psi(t)$ is an unknown bounded function. Then, the error dynamics could be acquired through Equations (13) and (17) as follows

$$\begin{cases} \dot{e}_1 = e_2 - \beta_1 e_1 \\ \cdots \\ \dot{e}_n = e_{n+1} - \beta_n e_1 \\ \dot{e}_{n+1} = -\beta_{n+1} e_1 - \psi(t) \end{cases} \tag{18}$$

The above formula can be rewritten as:

$$\dot{E} = HE + B(-\psi(t)) \tag{19}$$

where $E = Z - X$, $B = \begin{bmatrix} 0 & 0 & \cdots & 1 \end{bmatrix}^T$, the matrix $H$ is of Hurwitz type described as

$$H = \begin{bmatrix} -\beta_1 & 1 & \cdots & 0 \\ \vdots & & \ddots & \vdots \\ -\beta_n & & & 1 \\ -\beta_{n+1} & & \cdots & 0 \end{bmatrix} \tag{20}$$

Supposing the function $\psi(t)$ is bounded, that is $|\psi(t)| \leq \Delta$, then the following theorem is given:

**Theorem 1.** *Under above discussion, convergence condition for Equation (19) is* $\|E\| \leq \frac{2\Delta\lambda_{max}(P)}{\lambda}$.

**Proof of Theorem 1.** Define a Lyapunov positive function as $V = E^T PE$, then the time derivative of $V$ is as follows

$$\dot{V} = E^T \left( H^T P + PH \right) E + 2E^T PB(-\psi(t)) \leq -\lambda\|E\|^2 + 2\Delta\lambda_{max}(P)\|E\| \tag{21}$$

where $H^T P + PH = -\lambda I$, $I$ is unit matrix, $\lambda$ is the eigenvalue of positive definite symmetric matrix $P$. By choosing $\|E\| \leq \frac{2\Delta\lambda_{max}(P)}{\lambda}$, then $\dot{V} \leq 0$ is met.

Hence, the proof of convergence is completed.  □

### 3.3. State Error Feedback Control Law

After completing the extraction of error signals with each order, it is necessary to design a suitable control law to achieve the desired control goal of smooth and stable tracking. In this section, the state error feedback control law with n-order is designed as

$$u = \frac{u_0 - z_{n+1}}{b_0} \tag{22}$$

where $u_0 = \sum_{i=1}^{n} \varsigma_i(v_i - z_i)$, $\varsigma_i(i = 1, \cdots, n)$ are controller gains.

## 4. Stability Analysis

Although the ADRC has been successfully applied to industrial production, owing to its complex nonlinear characteristics, there lacks rigorous stability analysis for a long time. As we all known, stability provides a prerequisite for system to operate normally. Most of the stability analysis methods of ADRC are graphical in frequency domain [35,36]. In this study, the extended circle criteria are introduced to analyze the stability of dual-axis servo tracking turntable system. The conversion from the improved ADRC system to Lurie system is completed first.

### 4.1. System Transformation

Some assumptions need to be given before system transformation:

**Assumption 1.** *The input and all the outputs of improved TD is zero.*

**Assumption 2.** *The controlled plant described as Equation (17) is rewritten with a linear structure as follows*

$$\begin{cases} \dot{x}_1 = x_2 \\ \cdots \\ \dot{x}_n = a_n x_1 + a_{n-1} x_2 + \cdots + a_1 x_n + b_0 u \\ y = x_1 \end{cases} \tag{23}$$

*where $a_i(i = 1, \cdots, n)$ are the systematic parameters.*

Substitute Equations (12) and (22) into (23), then

$$\dot{x} = A_1 x + A_2 z + \eta z_{n+1} \tag{24}$$

$$\dot{z} = A_3 z + \varepsilon u', \dot{z}_{n+1} = \beta_{n+1} u' \tag{25}$$

where $x = [x_1, \cdots, x_n]^T$, $z = [z_1, \cdots, z_n]^T$, and $u' = -\varphi(e_1)$,

$$A_1 = \begin{bmatrix} 0 & 1 & \cdots & 0 \\ & \vdots & \ddots & \vdots \\ 0 & 0 & \cdots & 1 \\ a_n & a_{n-1} & \cdots & a_1 \end{bmatrix}, A_2 = \begin{bmatrix} 0 & \cdots & 0 \\ \vdots & \ddots & \vdots \\ 0 & \cdots & 0 \\ -\varsigma_1 & \cdots & -\varsigma_n \end{bmatrix}, A_3 = \begin{bmatrix} 0 & 1 & \cdots & 0 \\ & \vdots & \ddots & \vdots \\ 0 & 0 & \cdots & 1 \\ -\varsigma_1 & -\varsigma_2 & \cdots & -\varsigma_n \end{bmatrix}, \eta = \begin{bmatrix} 0 \\ \cdots \\ 0 \\ -1 \end{bmatrix},$$

$\varepsilon = \begin{bmatrix} \beta_1 & \cdots & \beta_n \end{bmatrix}^T$, $e_1 = c_1^T x + c_2^T z = \begin{bmatrix} -1 & 0 & \cdots & 0 \end{bmatrix} x + \begin{bmatrix} 1 & 0 & \cdots & 0 \end{bmatrix} z$.

Combined with the above formulas, we can obtain

$$\begin{cases} \dot{x} = A_1 x + A_2 z + \eta z_{n+1} \\ \dot{z} = A_3 z + \varepsilon u' \\ \dot{z}_{n+1} = \beta_{n+1} u' \\ e_1 = c_1^T x + c_2^T z \\ u' = -\varphi(e_1) \end{cases} \tag{26}$$

Equation (26) is a Lurie problem, proposed by Lurie in 1940s, which is an efficient method to analyze the stability of nonlinear system. The general structure of Lurie system is shown in Figure 7. The main idea of Lurie system is to transform a complex nonlinear system into the combination form of linear forward channel and nonlinear feedback structure. By analyzing the stability of linear forward part, the stability of original system is explained.

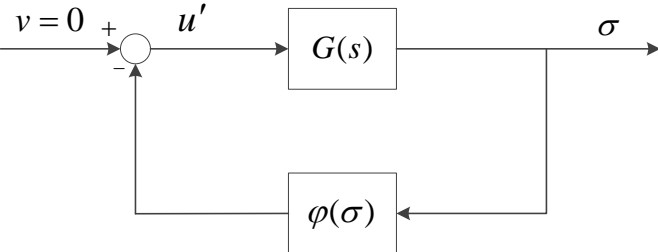

**Figure 7.** The general structure of Lurie system.

The original expression of Lurie system is as follows:

$$\begin{cases} \dot{x} = A x + \varrho u' \\ \dot{\xi} = u' \\ \sigma = c^T x + \rho \xi \\ u' = -\varphi(\sigma) \end{cases} \tag{27}$$

where $A = \begin{bmatrix} A_1 & A_1 A_2 \\ 0 & A_3 \end{bmatrix}$, $\varrho = \begin{bmatrix} \eta \beta_{n+1} \\ \varepsilon \end{bmatrix}$, $c^T = \begin{bmatrix} c_1^T A_1^{-1} & c_2^T \end{bmatrix}$, and $\rho = -c_1^T A_1^{-1} \eta \beta_{n+1} = -\frac{\beta_{n+1}}{a_n}$, then the linear forward part of Lurie system is described as

$$G(s) = c^T (sI - A)^{-1} \varrho + \frac{\rho}{s} \tag{28}$$

### 4.2. Extended Circle Criteria

After completing the system transformation, some theorems, see [33] for details, are introduced to analyze stability of the designed system.

**Definition 1.** *There exists circular region* $D(\mu_1, \mu_2)$, *whose diameter is the line segment connecting the points* $\frac{-1}{\mu_1} + j0$ *and* $\frac{-1}{\mu_2} + j0$.

**Theorem 2.** *Supposing that the transfer function of Lurie system described as Equation (28) has no eigenvalues on the imaginary axis except for m eigenvalues at the origin, and no eigenvalues with positive real parts at the same time, then two sufficient conditions must be met for global asymptotic stability: (i) as* $\omega$ *moves from* $0^-$ *to* $0^+$, *the Nyquist plot of* $G(j\omega)$ *circles m clockwise semicircles with infinite radius along the origin; (ii) the Nyquist plot of* $G(j\omega)$ *lies on the right of vertical line given as* $Re(s) = -\frac{1}{\mu_2}$.

**Remark 1.** *The above theorem gives the general criterion for the stability of Lurie system with some zero eigenvalue. In fact, there are other cases that may make system unstable, such as the existence of eigenvalues with positive real parts. Referring to the stability analysis methods of linear system in frequency domain, the extended circle criterion is given as follows.*

**Theorem 3.** *Supposing that the linear forward part of Lurie system (28) has no eigenvalues on the imaginary axis except for m eigenvalues at the origin, and p open-loop poles with positive real parts at the same time, then the necessary and sufficient condition for the stability of original system could be summarized as follows: (1) the first condition in Theorem 2 must be satisfied; (2)* $p = 2N$, *where N is the circles of Nyquist diagram of* $G(j\omega)$ *around the point* $(-1, j0)$.

### 4.3. Stability Analysis Based on the Extended Circle Criteria

The mathematical model using second order transfer function of turntable tracking servo system is introduced in Section 2, where the system parameters are shown in Table 1. As a result, Equation (3) can be described by the second order differential equation as follows:

$$\begin{cases} \dot{x}_1 = x_2 \\ \dot{x}_2 = f(x_1, x_2) + 28u = -2.38x_1 - 0.31x_2 + 28u \end{cases} \tag{29}$$

where $f(x_1, x_2) = -2.38x_1 - 0.31x_2$.

**Table 1.** Parameters of turntable tracking servo system.

| Parameters | Unit | Value |
|:---:|:---:|:---:|
| $J$ | kg·m$^2$ | 0.00272 |
| $K_t$ | N·m/A | 1.36 |
| $K_e$ | V/(rad/s) | 0.085 |
| $R_a$ | Ω | 5.65 |
| $L_a$ | H | 17.8 |
| $N$ | / | 112 |

In Section 3, we designed the improved ESO based on fuzzy algorithm. By repeated tuning, a group of desired parameters are listed in Table 2.

Then, the linear forward part of Lurie system in this study is as follows:

$$G(s) = \begin{bmatrix} 0.13 \\ 0.42 \\ 1 \\ 0 \end{bmatrix}^T \begin{bmatrix} s & -1 & 6 & 1.5 \\ 2.38 & s+0.31 & -1.86 & -0.46 \\ 0 & 0 & s & -1 \\ 0 & 0 & 6 & s+1.5 \end{bmatrix}^{-1} \begin{bmatrix} 0 \\ -1200 \\ 54 \\ 320 \end{bmatrix} + \frac{504}{s}$$

$$= \frac{54(s^2+2.413s+5.737)(s^2+5.318s+23.23)}{s(s^2+0.31s+2.38)(s^2+1.5s+6)} \tag{30}$$

**Table 2.** The desired parameters of Improved ADRC.

| Description | Parameters | Value |
|---|---|---|
| Improved ESO | $\alpha_1$ | 0.5 |
| | $\alpha_2$ | 0.25 |
| | $\alpha_3$ | 0.125 |
| | $\delta$ | 0.01 |
| | $\gamma$ | 0.1 |
| | $\beta_1$ | 54 |
| | $\beta_2$ | 320 |
| | $\beta_3$ | 1200 |
| SEF | $\varsigma_1$ | 6 |
| | $\varsigma_2$ | 1.5 |

Obviously, the transfer function $G(s)$ has no eigenvalues on the imaginary axis except for one eigenvalue at the origin. The open-loop poles are denoted as $p_1 = 0$, $p_{2,3} = -0.75 \pm j2.3318$ and $p_{4,5} = -0.15 \pm j1.5349$. For the purpose of verifying the robustness of the improved ADRC system further and to analyze the stability margins, the function $f(x_1, x_2)$ in Equation (29) is chosen as plus minus 20% of the true value for experiments. The Nyquist diagram and its local enlarged curve near the point $(-1, j0)$ are shown in Figure 8. The curve G1 represents the Nyquist plot of original system, while the curves G2 and G3 represent the counterpart with plus and minus 20% of the true value, respectively. For curves G1 and G2, the circle of Nyquist diagram around the point $(-1, j0)$ is $N = 2 - 2 = 0$, while it is $N = 1 - 1 = 0$ for curve G3. It could be concluded that the designed improved ADRC system is globally asymptotic stable according to Theorem 3. Results illustrate the stability and robustness of the proposed scheme.

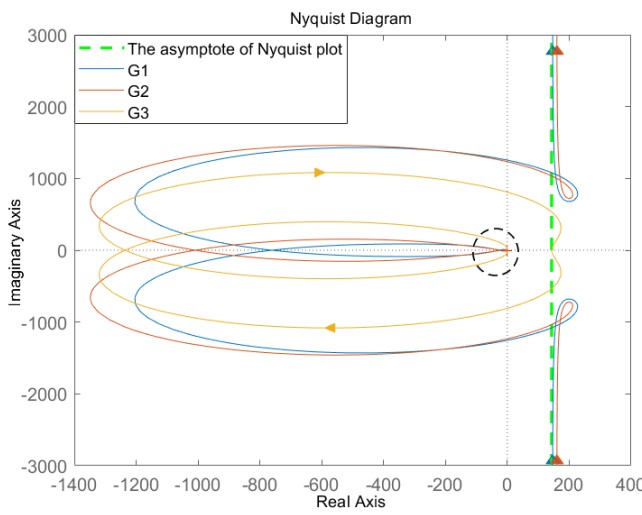

(**a**) The Nyquist diagram of Lurie system

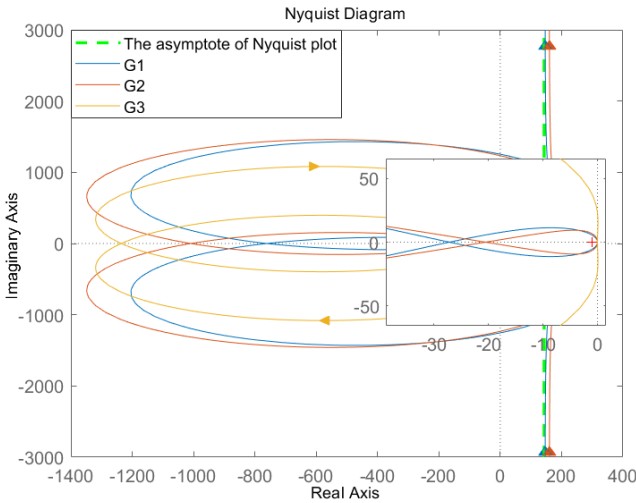

(**b**) The local enlarged curve of Nyquist diagram for Lurie system

**Figure 8.** The Nyquist diagram and its local enlarged curve of Lurie system.

## 5. Simulation Results and Analysis

After completing the stability analysis of turntable tracking servo system, some simulations are conducted to demonstrate the performance of improved ADRC scheme. The second order differential equation model of turntable tracking servo system is given as Equation (29) in Section 4. In order to compensate for the influence of nonlinear friction disturbance on the tracking performance better, a friction observer is designed based on the improved ESO, which regards the immeasurable part of EP friction model as a new

state. In our previous work, the nonlinear friction parameter identification experiments using genetic algorithm (GA) is analyzed in [23]. Similarly, in this study, the parameter identification results of EP friction model based on GA are listed in Table 3.

**Table 3.** Identification results of EP friction model.

| Parameters | Unit | Value |
|:---:|:---:|:---:|
| $T_c$ | N·m | 0.0274 |
| $T_s$ | N·m | 0.5724 |
| $v_s$ | rad/s | 0.0387 |
| $\sigma_0$ | / | 250 |
| $\sigma_1$ | / | 2.6 |
| $\sigma_2$ | / | 0.0291 |

In this section, a comparative simulation experiment based on improved ADRC, PD controller and traditional ADRC is analyzed to verify the tracking performance of the proposed control strategy. The same parameters are selected for the improved ADRC and traditional ADRC. We set the sinusoidal signal $y_d = \sin(t)$ (rad) as the expected trajectory signal. Figure 9 gives the estimated results and error curve of friction observer. The angle and angle velocity tracking result of servo turntable is shown in Figures 10 and 11, respectively. For the purpose of comparing tracking effects of different controllers further, corresponding tracking error indexes are given in Tables 4 and 5.

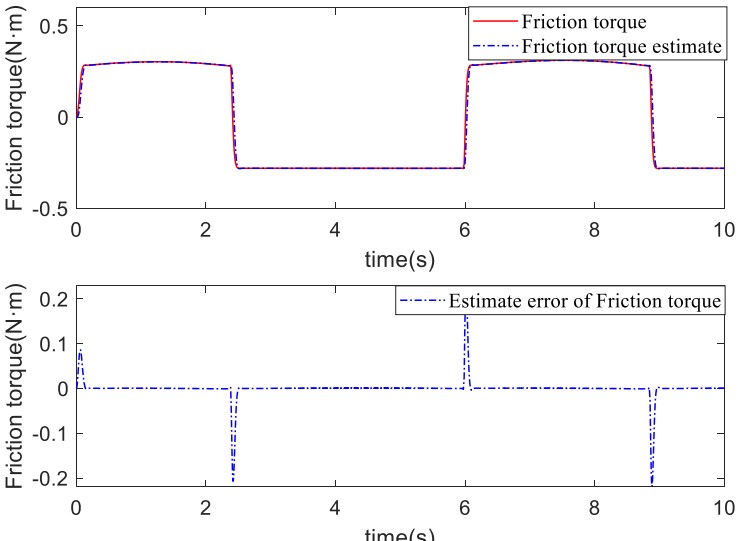

**Figure 9.** The friction torque estimate and error curve.

**Table 4.** The angle tracking error comparison.

| | MAX (rad) | MEAN (rad) | RMSE (rad) |
|:---:|:---:|:---:|:---:|
| PD controller | 0.5000 | 0.0145 | 0.1537 |
| ADRC | 0.3566 | 0.0015 | 0.0246 |
| Improved ADRC | 0.2000 | 0.00064 | 0.0161 |

**Table 5.** The angle velocity tracking error comparison.

| | MAX (rad/s) | MEAN (rad/s) | RMSE (rad/s) |
|:---:|:---:|:---:|:---:|
| PD controller | 5.4577 | 0.5464 | 1.4957 |
| ADRC | 4.3488 | 0.0414 | 0.3305 |
| Improved ADRC | 1.9509 | 0.0043 | 0.1081 |

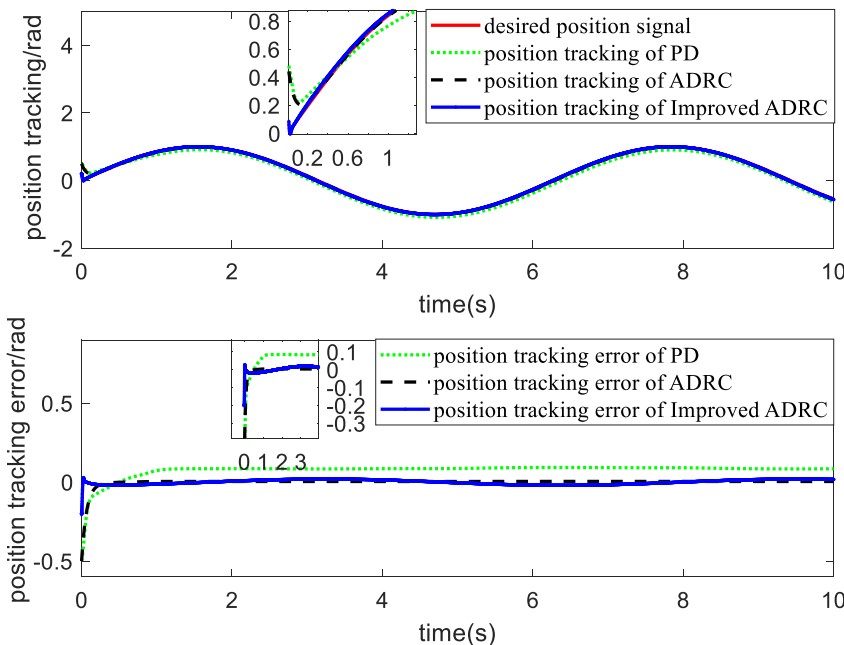

**Figure 10.** The angle tracking results and error curve.

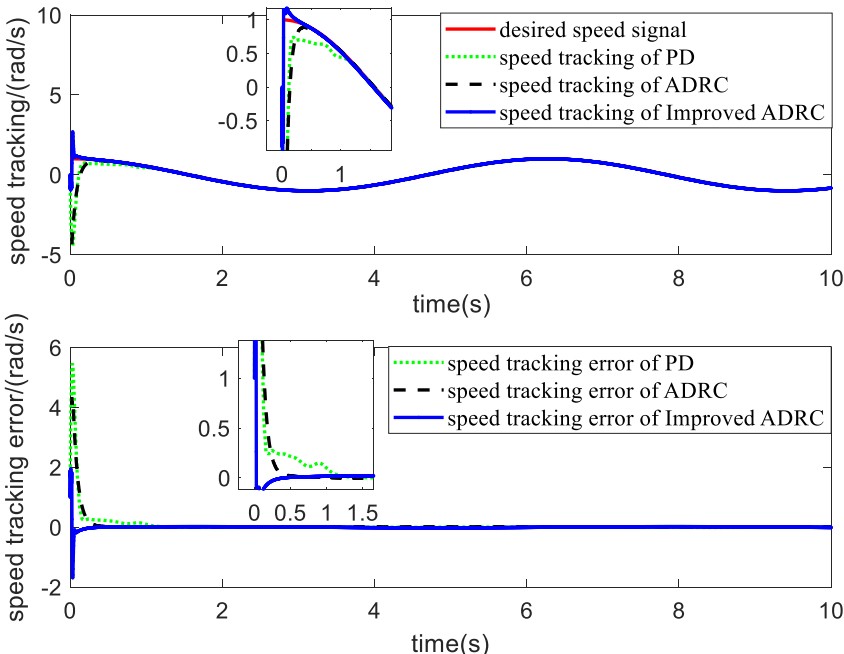

**Figure 11.** The angle velocity tracking results and error curve.

As is shown from the above tables, the maximum angle velocity tracking error of improved ADRC is 3.5068 rad/s less than that of PD control method and 2.3979 rad/s less than that of traditional ADRC. The MEAN of angle tracking error of improved ADRC is 0.00064 rad, which is far lower than the corresponding values of PD and ADRC. The root mean square error (RMSE) of improved ADRC is also with the lowest value. It is not hard to find out that the improved ADRC is with better performance and stronger robustness than PD and traditional ADRC.

Finally, for the purpose of verifying the effectiveness of the improved ADRC against parameters uncertainties further, numerical simulation experiments are carried out for unknown parameters. The function $f(x_1, x_2)$ in Equation (29) is unknown here, which is chosen as plus minus 20% of the true value for experiments. The test signals are selected

as $y_{d1} = 1.0 \cdot sin(t)$ (rad) and $y_{d2} = 1.0 \cdot sign(sin(t))$ (rad), respectively. Simulation results are shown in Figure 12, which illustrate the robustness and effectiveness of the proposed scheme.

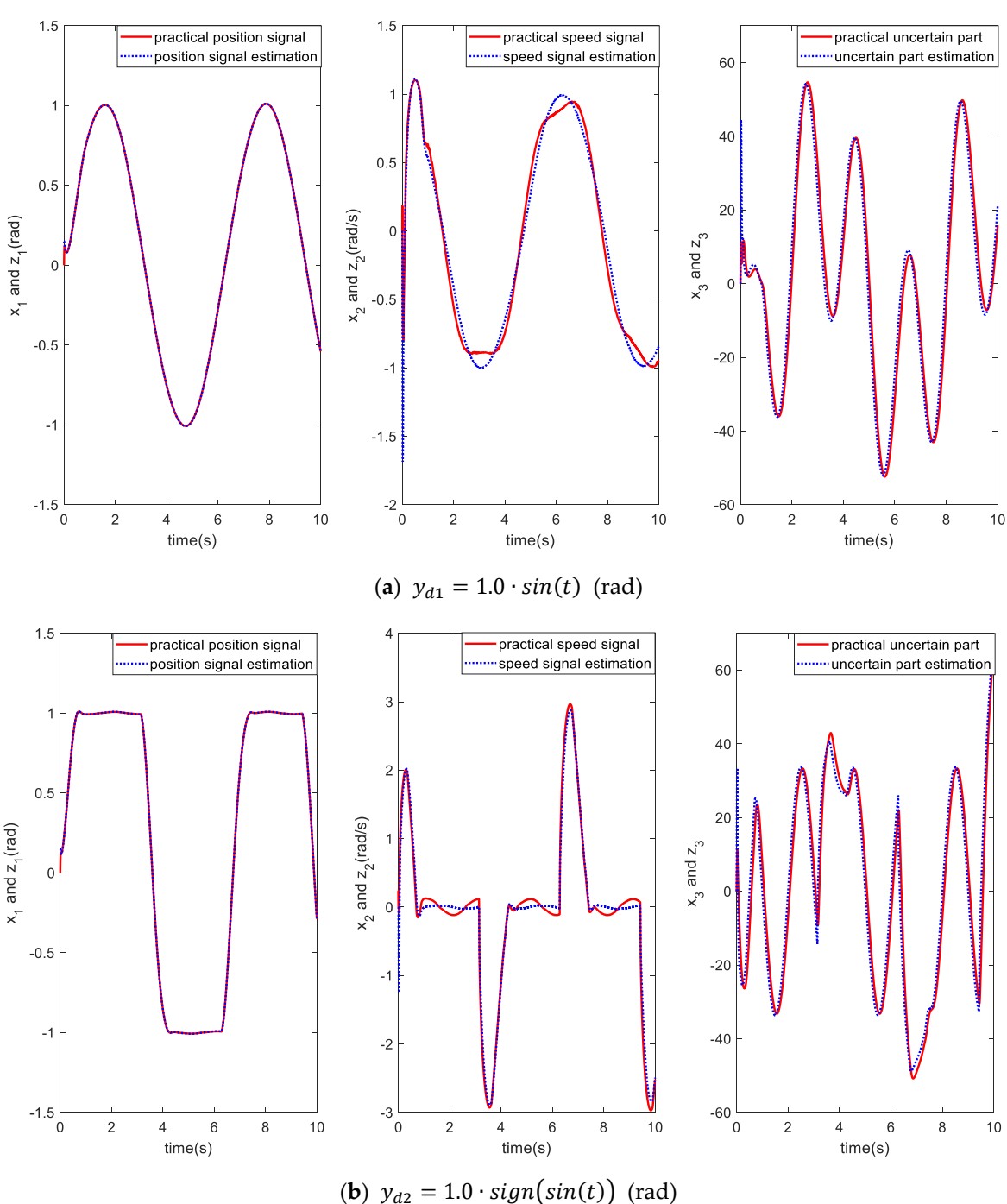

(**a**) $y_{d1} = 1.0 \cdot sin(t)$ (rad)

(**b**) $y_{d2} = 1.0 \cdot sign(sin(t))$ (rad)

**Figure 12.** The results of numerical verification experiments under different test signals.

## 6. Experimental Result

### 6.1. Experimental Platform

The experimental setup of dual-axis servo tracking turntable, as shown in Figure 13, is built to further verify the effectiveness of improved ADRC scheme in improving the tracking performance of servo system. In this section, an improved ADRC model designed above is established in Rtunit complier environment, which is given in Figure 14.

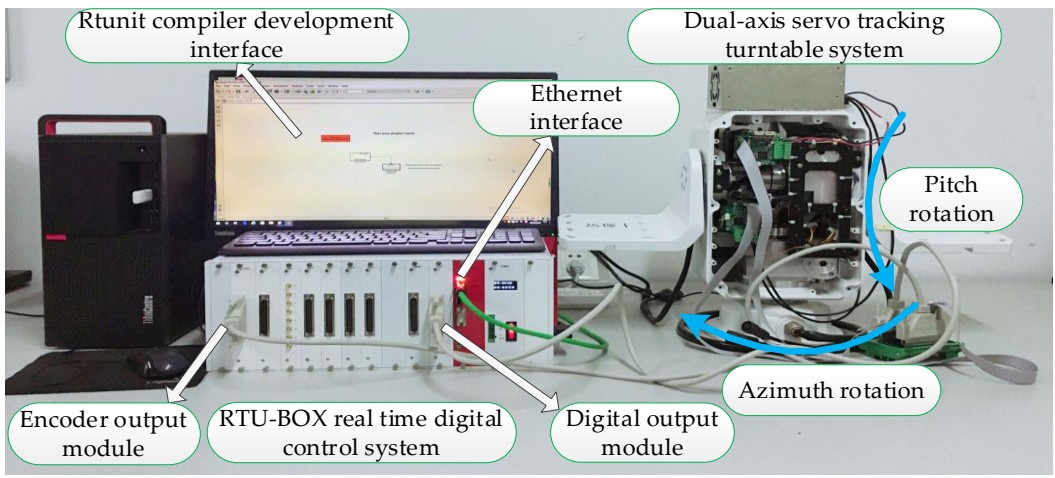

**Figure 13.** Experimental setup of dual-axis servo tracking turntable.

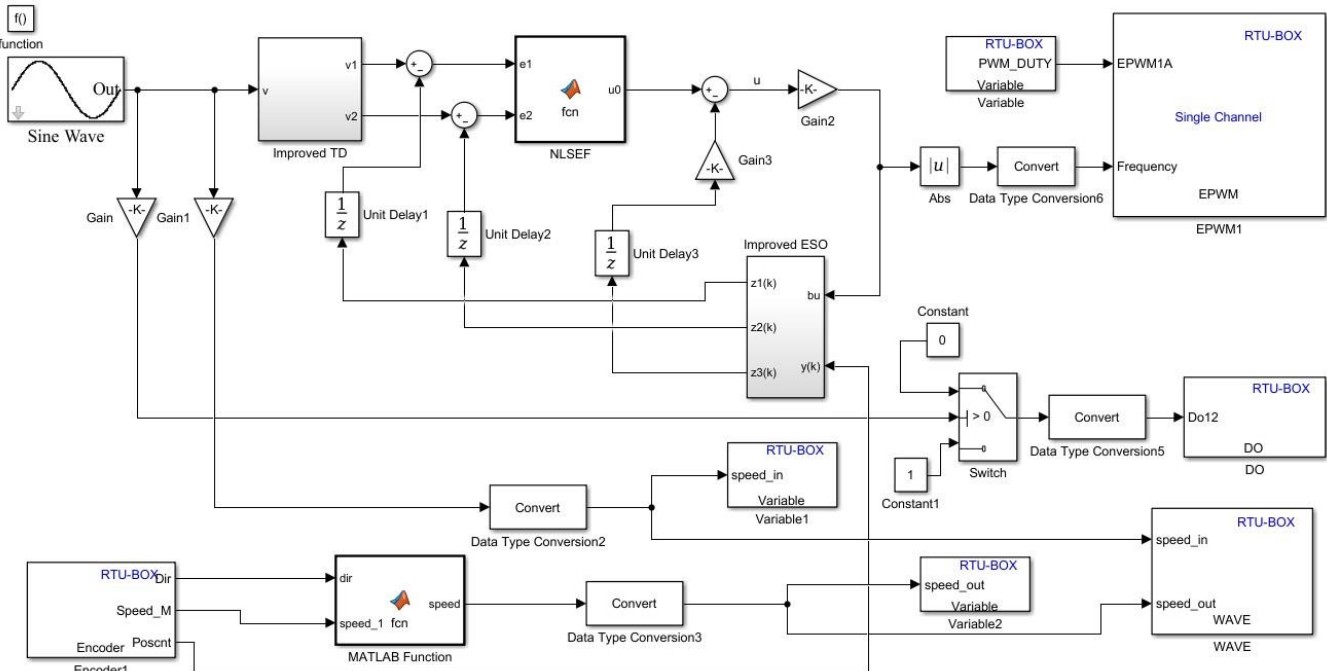

**Figure 14.** The improved ADRC model based on Rtunit complier environment.

The RTU-BOX real-time digital control system is mainly composed of Encoder Output module, Digital Output (DO) module, PWM module, etc. After completing the code generation and compilation of control program in the Rtunit compiler development interface, the host computer would download the compiled program to RTU-BOX through the Ethernet interface. Then, the control signal generated by DO module is arranged to drive servo turntable to rotate at a certain speed. The Encoder Output module can calculate and output the rotating speed of servo turntable in real time, with the resolution of 4096. The transmission ration of servo motor to the tracking turntable is 112:1. For the purpose of realizing dual-axis rotation, the DO12 and DO13 pins are employed to control the azimuth rotation and pitch rotation, respectively.

### 6.2. Experimental Results and Analysis

The dual-axis tracking turntable shown in Figure 13 is controlled by frequency signal. Figure 15 gives the rotational speed value of servo turntable tracking system under different constant frequency. We set a sinusoidal signal as input frequency, that is

$f = 3500 \cdot \sin(2\pi \cdot 50t)$ Hz, then the rotation speed tracking results of azimuth axis and pitch axis are shown in Figures 16 and 17, respectively. The blue line represents the tracking curve of traditional ADRC, and the green line denotes the counterpart curve of improved ADRC. Obviously, there exists distortion and chattering phenomenon in the peak and zero crossing regions based on traditional ADRC, while the improved ADRC can restrain these deficiencies to a certain extent. In order to make the different tracking effects of the two controllers more visual, Figures 18 and 19 give the rotation speed tracking error curves of azimuth axis and pitch axis in polar coordinates, respectively.

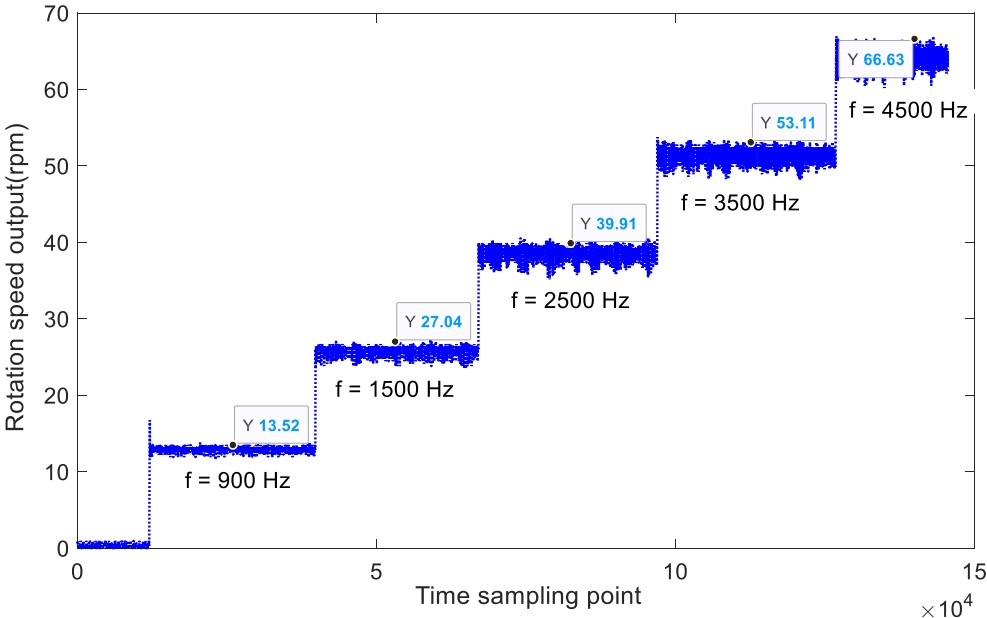

**Figure 15.** The rotational speed value under different constant frequency.

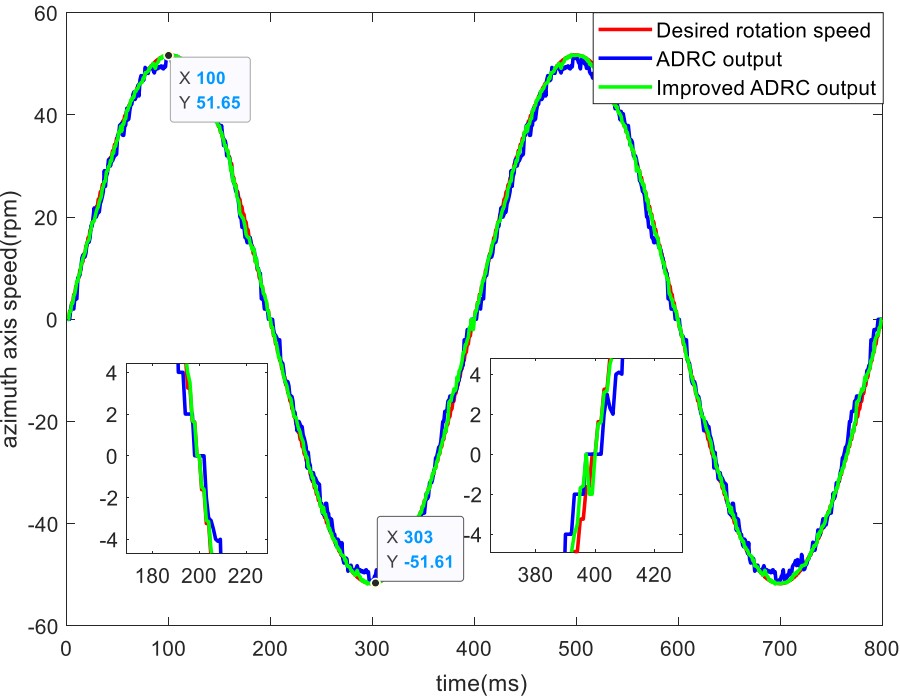

**Figure 16.** The rotation speed tracking results of azimuth axis.

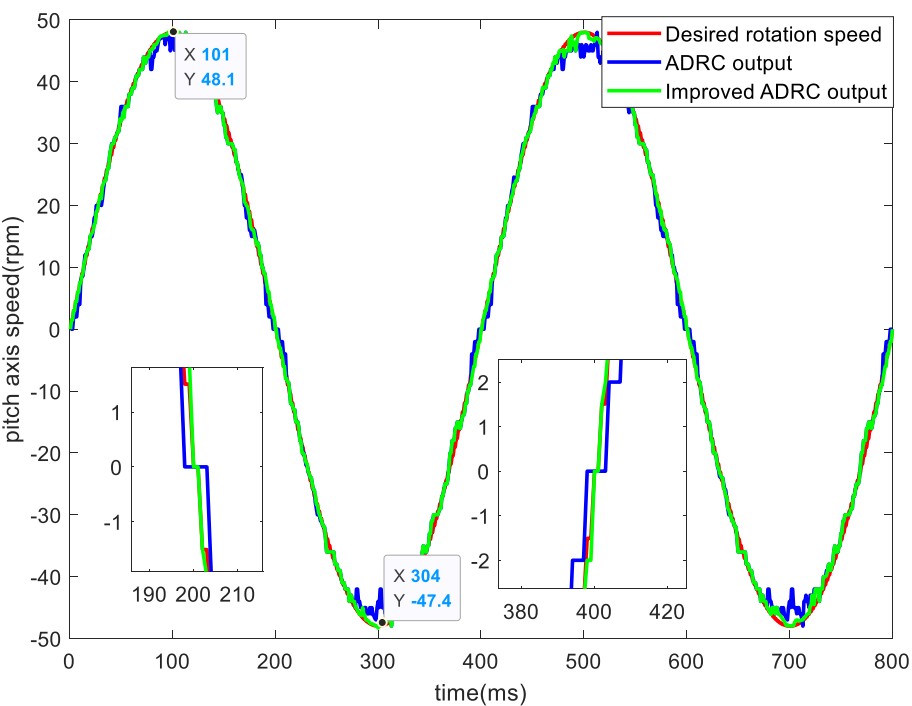

**Figure 17.** The rotation speed tracking results of pitch axis.

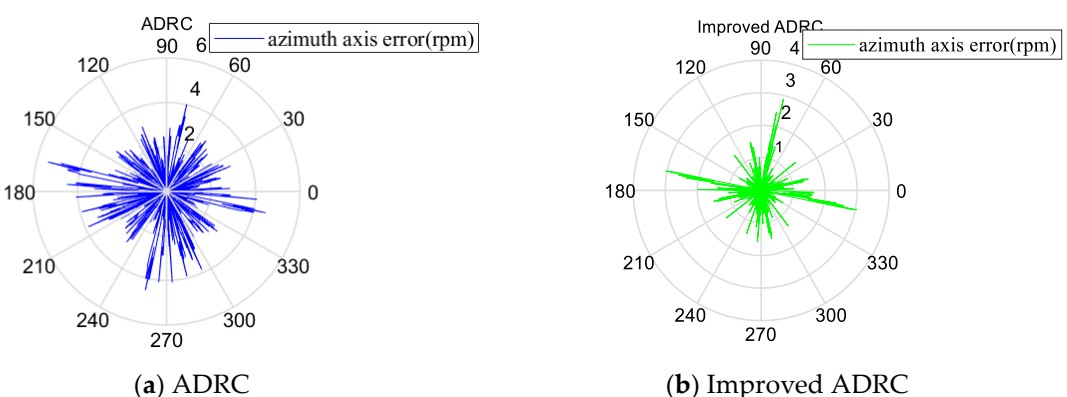

(**a**) ADRC          (**b**) Improved ADRC

**Figure 18.** The rotation speed tracking error of azimuth axis based on ADRC and Improved ADRC.

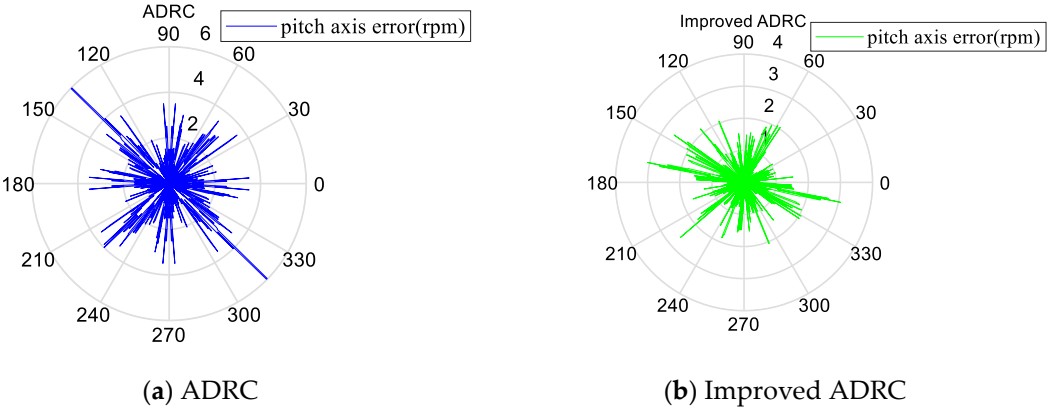

(**a**) ADRC          (**b**) Improved ADRC

**Figure 19.** The rotation speed tracking error of pitch axis based on ADRC and improved ADRC.

Finally, some error indexes are listed in Table 6 to further compare the tracking performance of the above two controllers. It can be found that maximum rotation tracking

error of improved ADRC is 1.9598 rpm and 3.1611 rpm less than that of traditional ADRC for azimuth and pitch rotation, respectively. The MEAN and RMSE of improved ADRC are also much less than traditional ADRC. By comparison with traditional ADRC, the improved ADRC can achieve with the more accurately tracking of servo turntable both azimuth and pitch rotation.

**Table 6.** The rotation speed tracking error comparison.

|  |  | MAX (rpm) | MEAN (rpm) | RMSE (rpm) |
|---|---|---|---|---|
| azimuth axis | ADRC | 4.8421 | 0.0669 | 1.6981 |
|  | improved ADRC | 2.8823 | 0.0300 | 0.5987 |
| pitch axis | ADRC | 5.9826 | 0.0526 | 1.5666 |
|  | improved ADRC | 2.8215 | 0.0463 | 0.8113 |

## 7. Conclusions

In this study, an improved ADRC, which applies the improved TD and improved ESO, is proposed to realize high-precision tracking control of dual-axis servo tracking turntable. The mathematical model is established first, and the Elastoplastic (EP) friction model is adopted to describe friction nonlinear disturbance. Secondly, considering the properties of smooth and monotonically increasing, an improved TD is given based on hyperbolic tangent function. Thirdly, the improved ESO is analyzed in detail, which adopts a new nonlinear function. The immeasurable part of EP friction model is extended to a new state of improved ESO design. Additionally, the fuzzy algorithm is employed to tuning observer gains intelligently. Finally, the convergence of improved ESO is confirmed, and the improved ADRC system is transformed into a Lurie system, the stability of system is analyzed by extended circle criteria. It can be concluded that the maximum rotation tracking error of improved ADRC is 1.9598 rpm and 3.1611 rpm less than that of traditional ADRC for azimuth and pitch rotation, respectively. Simulation and experiment results demonstrate the effectiveness and robustness of the proposed control scheme.

To further improve the tracking and anti-disturbance performance of the turntable servo system in the future, one can focus on the compensation of other nonlinear disturbances, such as backlash, dead-zone and motor torque fluctuation, etc.

**Author Contributions:** Conceptualization, Q.Z. and X.W.; methodology, X.W.; software, C.Y.; validation, Q.Z., X.W. and Q.W.; formal analysis, C.Y.; investigation, X.W.; resources, Q.Z.; data curation, X.W.; writing—original draft preparation, X.W.; writing—review and editing, X.W.; visualization, D.C.; supervision, Q.Z.; project administration, D.C.; funding acquisition, Q.W. All authors have read and agreed to the published version of the manuscript.

**Funding:** This research was funded by National Natural Science Foundations (NNSF) of China under Grant 51637001 and 52077001, post-doctoral research foundation of Anhui Province Z01011804.

**Conflicts of Interest:** The authors declare no conflict of interest.

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
