# Peer review of "Improved Active Disturbance Rejection Control of Dual-Axis Servo Tracking Turntable with Friction Observer"

_electronics, doi:10.3390/electronics10162012_

Round 1
Reviewer 1 Report
The manuscript proposes an improved active disturbance rejection control scheme of dual-axis servo turntable with experimental verification. Hence, the idea is interesting and sounded, however, there are some issues that should be addressed by the authors:
1) The paper suffers from some language challenges. The paper should be proofread by a native speaker or a proofreading agent. Please, revise the readability and presentation well to be simple to the reader.
2) The "Abstract" and "Introduction" sections should be made much more impressive by highlighting your contributions. The novelty of the proposed method must be explained simply and clearly in points at the end of the introduction section.
3) the authors should add another paragraph to enrich the literature survey part by speaking about the stability, nonlinearities, and robustness of PID controller designation background and fuzzy control by adding and citing the latest up-to-date references 2021, e.g., Effective Nonlinear Model Predictive Control Scheme Tuned by Improved NN for Robotic Manipulators & Resilient Design of Robust Multi-Objectives PID Controllers for Automatic Voltage Regulators: D-Decomposition Approach (DOI: 10.1109/ACCESS.2021.3100415) & Robust Design of ANFIS-based Blade Pitch Controller for Wind Energy Conversion Systems.
4) It must check all the citing references of equations (1): (30), there are some missing ones. In addition, check again carefully all the abbreviations definitions, or symbols of all equations in the whole manuscript.
5) The performance of the proposed controller method should be better analyzed, commented and visualized in the experimental section. Also, the authors should tabulate the controller parameters and their comparison with another traditional controller supported with more discussions.
6) The authors should study the robustness test to check the effectiveness of the proposed controller against parameters uncertainties.
7) The conclusion section is weak; it should be rearranged, and numerical results should be added. According to the proposed controller, the authors may propose some interesting problems as future work in the conclusion.
Author Response
Thanks a lot for your advice! We studied the Reviewer Comments and revised our manuscript following the comments. All the questions are answered sentence by sentence, and the revisions are marked red in the manuscript.

Reviewer 2 Report
The authors present an ADRC approach to nonlinear friction compensation effects in a dual-axis turntable to improve tracking performance.
At first the authors present the areas of application, screening the existing literature. The authors claim to improve the ADRC's performance by introduction of a tracking differentiator and by improving the state observer.
Subsequently, the authors identify factors which impede good tracking perforance, among which the already-mentioned friction is present. On the contrary to the exising approaches, the authors include friction-related info to be a part of the state vector estimated by the extended state observer.
The authors have identified the main contributions of their study, however, they use a strange naming, as Lur'e systems are denoted as Lurie. They also omit independent variable information in (1) or (2), for example Te instead of Te(t), but they use Theta(s) for instance. This impedes the tracking of the ideas of the authors.
Having presented the ADRC scheme, the tracking differentiator is presented. The question is why not to use a fractional-order differentiator instead of a second-order like behaviour subject to proper selection of adjustable parameters? Can this be generalized?
in order to use a LESO approach, the appropriate model of the system should be known. Can you present any results subject to mismodelling errors, such as uncertain gain, or uncertain time constant (plus minus 20% of the true value to show the robustness of your method)?
Can you possibly deliberate on the possiblity to use a feedback linearization approach to the structure of the system from Fig. 7 to obtain a linear system? Where would the possible bottleneck lie? Can you linearize the nonlinear element by adopting the same tan-related approach?
Can you present the already-mentioned robustness results on a Nyquist plot such as in Figure 8 for border values?
The paper is promising, presents experimental results, which would definitely look much better should the authors include also model mismatching to present experimental results.
Author Response

(The authors gave the same response as above.)

Round 2
Reviewer 1 Report
Thanks for your modifications.
Author Response
Thank you very much for your affirmation after I revised my manuscript under your advice. On behalf of my co-authors, we would like to express our great appreciation to the editor and reviewers.
Reviewer 2 Report
Thank you for taking my comments into consideration, nevertheless I find some of points not addressed accurately.
Point 3 of my prior review - you have included Figure 12 in your new manuscript, but there is no information concerning uncertainty of the parameters - is there any mismatch between the plant and the model? If so, of what type and with what parameters? How do you estimate parameters? What about my suggestion of inclusion of 20% uncertainty?
Point 4 - I believe I have expressed myself clearly enough- one of the strands to control a nonlinear plant is to linearize it through the FBL method. After that, you only get a linear plant to control. Please see 10.1007/978-1-4612-1416-8_3 as an example.
Point 5 - so what if you included wrong parameters instead of the true ones? The Nyquist plot would definitely change leading to decrease of stability margins. This needs some comments and visualisation.
Author Response

(The authors gave the same response as above.)

Round 3
Reviewer 2 Report
Thank you for taking my comments into consideration. I fully agree with your response, though as FBL is as you have written - challenging - it does not mean it presents no solution to the problem! As far as the DOI is connected, typing it into Goole shows Feedback Linearization | SpringerLink. Good luck with the reviewing process.